environmental science/energy/mechanical engineering

shallow depth seam, highwall mining, rock fall, coal pillar stability, remote control, safe and efficient mining

**Author for correspondence:**
Cun Zhang
e-mail: cumt_zc@163.com

# Reasonable coal pillar design and remote control mining technology for highwall residual coal resources

## Fangtian Wang[1] and Cun Zhang[2]

[1]Key Laboratory of Deep Coal Resource Mining (CUMT), Ministry of Education of China, School of Mines, China University of Mining & Technology, Xuzhou 221116, People's Republic of China
[2]School of Resource and Safety Engineering, State Key Laboratory of Coal Resources and Safe Mining, China University of Mining and Technology (Beijing), Beijing 100083, People's Republic of China

CZ, 0000-0001-8673-3077

Highwall mining (HWM) technology is an efficient method for exploiting residual coal resources in Chinese open-pit coal mines. However, on-site personnel and equipment can be damaged by the instability of the highwall mining residual coal pillars and subsidence of final end-walls. This paper considers the geological conditions of an open-pit mine in Shendong Coal Field (China) in order to prevent overlying rock fall accidents; the Mark-Bieniawski formula and the FLAC3D numerical simulation are used to analyse reasonable coal pillar widths outside and under the road, which were determined to be 1.7 m and 1.3 m, respectively. Using the EBH132 cantilever excavator for remote control mining, the field experiment shows that the recovery ratio of highwall residual coal resources was over 67%; hence, safety, efficiency and high recovery ratio of highwall mining were achieved for the residual coal resources of an open-pit mine.

## 1. Introduction

The slope angle, mining boundary, changes of the coal seam thickness, etc. are the main reasons that many coal resources remain under the end-walls in open-pit coal mines [1–3]. Because of the undeveloped mining technology and low economic benefits, those resources were discarded or exploited using the room and pillar method with low recovery ratios. This issue causes waste of resources and introduces safety risks such as the spontaneous combustion of the coal seam, surface vegetation destruction, rock fall accidents, slope landslides and large-area collapse because of the instability of the residual

pillars. Since highwall mining has the advantages of a high recovery ratio, easy manoeuvrability, safety and low production cost, states and enterprises have paid increasing attention to this method [4–6]. There is valuable and challenging research on how to recycle the residual coal under the end-walls safely and efficiently [7,8].

Highwall mining is a technique to obtain additional coal recovery after the economic strip limit is reached in surface mining. It involves the remote deployment of a continuous miner in openings beneath the final highwall with no personnel entry. Many candidate areas for highwall mining have thick and steeply dipping seams. Mining down dip presents challenges related to the maximum pulling capacity of the machine, traction of the cutting head, and material conveying, all of which limit the penetration depth. The maximum penetration is greater for flatter slopes and decreases for slopes near the threshold of the maximum machine operating angle. Most highwall mining operations are relatively flat with slight undulations in the seam; therefore, the highwall mining pillar design criteria apply fairly equally to the entire mining area. However, in steeply dipping deposits, design criteria based on higher overburden loads at the far end of the penetration are excessively conservative for the shallower portions of the openings near the highwall.

To recover coal remnants around the end-walls, an underground mining system is normally adopted by excavating some adits into the end-walls [9–11]. However, because of the small area of residual coal resource and poor production conditions, it is not beneficial for the layout of the traditional longwall working face. If the full caving method is used to manage the coal roof, it will cause a larger ground subsidence, which is not conducive to the stability of the slope. Zonal mining is commonly used in regions where only minor ground placement is permissible. With this method, the main haulage and ventilation roadways are designed and excavated from the exposed position of the coal seam in the end-walls. For the irregular and small area of an end-wall residual resource, the costs remain high [12–14]. Thus, it is difficult to exploit small-area residual coal resources with conventional techniques, but remote control technology can be a good solution to this problem [15,16]. Many countries throughout the world have been investigating remote control coal mining technologies and their equipment, which is shown in figure 1. This technology has been developed after much field practice [17–19]. Moreover, the mining parameters significantly affect the stability of the end-wall in the mining process of the residual coal resource. Hence, to ensure the stability of the end-wall and minimum sinkage for the upper highway, the mining parameters must be analysed.

A reasonable mining pillar for highwall mining is conducive to safe and efficient mining. The highwall mining pillar design is a direct function of the coal strength, opening height, opening width, and depth of cover. An elasto-plastic model suitable for the analysis of coal pillars has been developed and implemented in both two- and three-dimensional finite-element codes by Fama et al. [20]. The use of the local mine stiffness concept can provide added confidence in a highwall mining panel layout design [21]. Web and barrier pillar recommendations for close-proximity multiple-seam highwall mining were studied by Mark [22,23]. Perry et al. [24] studied the effect of the highwall mining progression on the web and barrier pillar stability. Using numerical modelling tools, a correction factor was suggested in the empirical pillar strength equation for slender pillars with width-to-height ratios less than unity [25]. However, current studies rarely consider the effect of the roads on the highwall and pillar design. The effect of the coal trucks on the road on the stability of coal pillars was also of less consideration.

Both theoretical analysis and numerical simulation are used in this paper to calculate the reasonable width of pillars based on the background of highwall mining in a coal mine. The EBH132 cantilever excavator was used for the remote control mining. The safe, efficient and high-recovery-ratio highwall mining was achieved for the residual coal resources of an open-pit mine.

## 2. Engineering background

The study coal mine is located in Inner Mongolia Autonomous Province, China. Its northwestern boundary borders on another coal mine, and the southwestern boundary is linked to a highway. There is a 50–80 m wide, 390 m long reservation coal resource in the northwestern boundary due to the design requirement for the security pillar, boundary stage and road for transportation above the end slope. In the southwestern boundary, there are highways, 11-kVA high-voltage power lines, a 35-kVA electrical substation, boundary stage and road for transportation above the end slope, which resulted in the 180 m wide residual coal resources. It is difficult to exploit these coal resources using only conventional mining techniques. Thus, highwall mining technology was used in this paper to excavate the residual coal resource and improve the recovery rate.

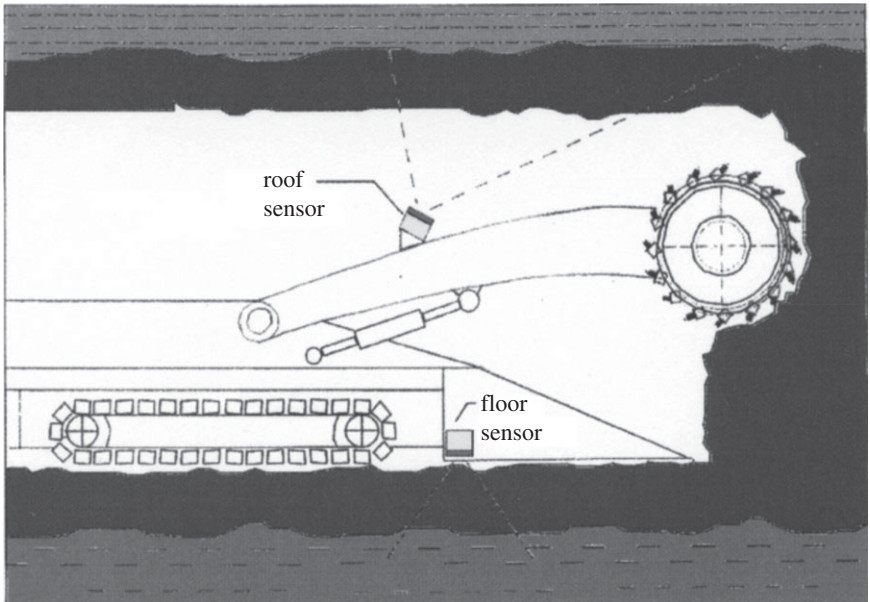

**Figure 1.** Horizon control problem. (Here, the underlying objective is to keep the mining machine in the coal seam to maximize the coal recovery. The black bands represent coal, and the textured zones represent the tuff.)

The area of highwall mining is located in the southwestern boundary of the 2# coal seam open-pit. It was laid out along the road and 145–240 m away from the road. The surface of the mining area is the sand dunes, whose ground elevation is 1300–1330 m, with no ground water or building. The slope angle is 45°, and the detail of the topographic map is shown in figure 2.

The working face is located in the 2# coal seam, and the average thickness is 3.5 m. The dip angle is 1–2°, and the density is 1.4 t m$^{-3}$. The coal seam is simple with no dirt band. The roof is sand and fine sand mud; the interbed and floor are mainly sandy mudstone with partly argillaceous siltstone.

The main problem of the highwall mining technology is the layout of roadways. In this paper, remote control technology is used, and the excavate width is determined by the size of the EBH132 cantilever excavator. There is no need to extract the roadway for humans, which can simplify the mining system to achieve efficient mining. However, there is no permanent support in the roadway for the remote control mining. Thus, the stability of the overburden structure mainly depends on the stability of the retained coal pillars, and a reasonable pillar width ensures the safety of mining and increases the recovery ratio. Thus, the reasonable coal pillar design and remote control technology for the HWM technology were studied in this paper.

# 3. Reasonable coal pillar design to prevent rock fall accidents

## 3.1. Numerical simulation

### 3.1.1. Working status of the coal pillar

The working status of the coal pillar mainly includes three different situations, as shown in figure 3. Mohr–Coulomb failure criteria are used in the numerical model. A more plastic zone in the pillar indicates a lower bearing strength of the pillar. However, the remaining plastic zone has the residual strength to support the elastic zone, as shown in figure 3b. When the plastic zone cuts through the pillar, the plastic penetration area does not have sufficient strength to support the roadway, as shown in figure 3c. Thus, the safety conditions of the pillar are that the plastic zones do not cut through the pillar. Figure 3a,b satisfies the requirement. However, if the coal pillar width is too large as figure 3a shows, it will waste the resource and reduce the resource recovery. In general, the reasonable width of coal pillar is shown in figure 3b: exploit as much coal resource as possible under the safety condition of the pillars.

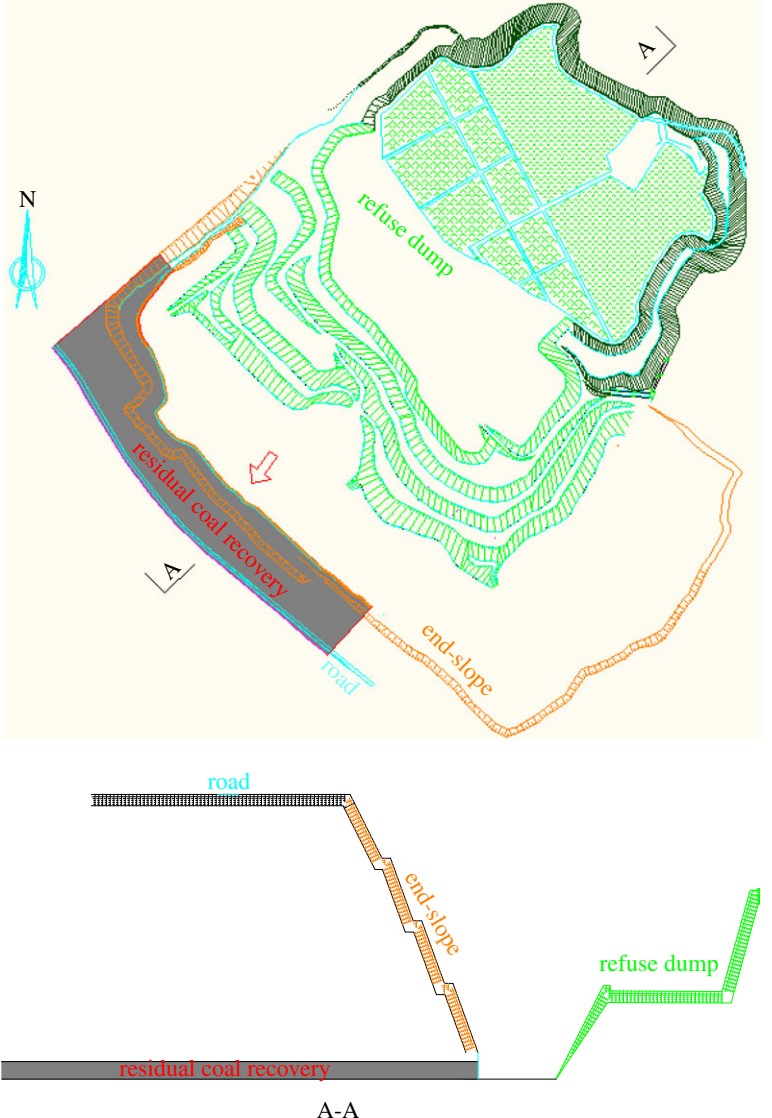

**Figure 2.** Topographic map of the highwall area.

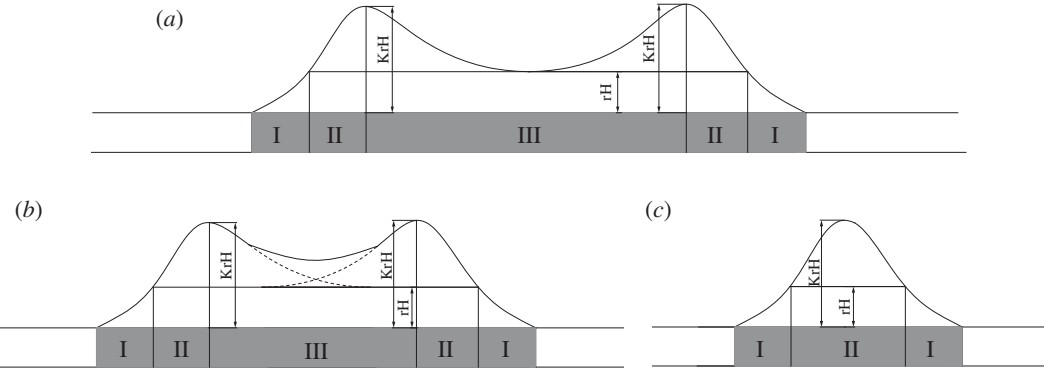

**Figure 3.** Plastic zone with different coal pillar widths. (*a*) Too large coal pillar width, (*b*) appropriate coal pillar width and (*c*) too small coal pillar width. I—fractured zone; II—plastic zone; III—elastic zone.

### 3.1.2. Simulation model parameters

To accurately simulate the deformation and failure characteristics of coal and rock in coal mining, the paper introduces the software FLAC3D, which is based on the finite difference method, to establish

the model. In this model, the rock stratum is represented by Mohr–Coulomb model, the coal seam is described by the strain softening model, the cohesion and friction angle of the coal seam degrades as the plastic shear strain increases, and these factors are assigned the residual values when the plastic shear strain reaches 0.01; the physical and mechanical parameters of the coal and rock in the model are listed in table 1. The numerical model and selected properties were calibrated through comparison using the coal uniaxial compressive strength test [26]. Figure 4 shows a good consistency between the numerical results and the laboratory test for the stress–strain curve and failure mode of the sample. The maximum shear strain in the numerical model shows an X-shaped failure mode of the sample, which is consistent with the laboratory test.

The pillar under the road outside was simulated to be 1.0 m, 1.3 m and 1.5 m wide, and the pillar under the road was simulated to be 1.4 m, 1.7 m and 2.0 m wide, respectively. According to the size of the residual coal and cantilever excavator, the width of each panel (sum of the pillar width and excavated width) was set to 5.5 m. Thus, the excavation width under the road outside was simulated to be 4.5 m, 4.2 m and 4 m, and the excavation width under the road was simulated to be 4.1 m, 3.8 m and 3.5 m, respectively. Both the ends and the bottom of the model were fixed. The parameters of all rocks are shown in table 1, and the diagram of the model is shown in figure 5.

To analyse the effect of the coal trucks on the road on the stability of the coal pillars, a dynamic load was applied in the model, as shown in figure 6. The load is assumed to be 0.01 MPa intervals of 1000 steps, since the load of a truck loaded with coal is approximately 0.01 MPa, and the dynamic balancing steps are 1000.

### 3.1.3. Simulation results analysis

Because of the shallow depth of the coal seam, the vertical stress of the coal pillar is normally less than its compressive strength, so the failure mode of the coal pillar conforms to Mohr–Coulomb yielding criteria. When plastic failure occurs in the coal pillar, the ultimate bearing capacity of the coal pillar declines. Thus, the strain softening model is used for the pillar, where the internal friction angle, cohesion and strength of extension decrease with the increase in strain. When the plastic zone spreads throughout the coal pillar, the ultimate bearing capacity will significantly decline and make the coal pillar unstable. The diagrams of plastic zones only show the partial cross-sections of coal pillars, and each grid that corresponds to the actual length is 0.1 m. Figures 7 and 8 show the plastic zone development of the coal pillar outside and under the road, respectively, and figure 9 shows the vertical displacement cloud diagram of the road cross-section.

As shown in figure 7, when the coal pillar outside the highway is 1.0 m, the plastic zone has spread throughout the entire pillar, and the coal pillar collapses because of its instability, which cannot satisfy the mining safety requirement; comparing the condition of 1.0 m, all plastic zones of 1.3 and 1.5 m decreased. Due to the corners of the pillar experiencing the highest stress concentration, the plastic zone priority occurs in this area. Thus, figure 7c with stress concentrated only at the corners shows that the bearing capacity can satisfy the requirement. While in figure 7b, after the corners of the pillar yielded, the plastic zone then occurred in the core of pillar, due to the brittleness of coal; when the stress reaches a certain strength, tensile failure will occur in the middle part, but the plastic penetration area did not appear, so both zones maintained the stability. Therefore, the width of the coal pillar under the road outside is 1.3 m due to the high resource recovery ratio.

As shown in figure 8, the plastic zone spreads throughout the entire coal pillar when the width under the road was 1.4 m. As shown in figure 9a, the bearing capacity of the coal pillar significantly decreased and caused the shrinkage to reach 16 cm, which is significantly more than the others. This result demonstrates that the coal pillar collapsed due to unstability. When the width increases to 1.7 m, the plastic zone is mainly distributed in the corners, but the scope is relatively small and does not spread throughout the whole pillar. The bearing capacity can satisfy the requirement. Meanwhile, the maximum shrinkage is 4.8 cm in the cross-section of the highway, which satisfies the engineering requirement. When the width reaches 2.0 m, the scope plastic zone further declines, and the maximum shrinkage is only 1.8 cm, which indicates that the coal pillar is stable. Comparing both conditions and considering the recovery rates, we obtain that the width of the pillar under the road is 1.7 m, which is 0.2 m larger on both sides than the pillar under the road outside.

The simulation result indicates that the reasonable widths of the pillar outside and under the road are 1.3 m and 1.7 m, respectively, which can satisfy the requirements for mining safety and a high recovery ratio. Therefore, this condition was considered the reasonable width of a coal pillar after comprehensive comparison.

**Table 1.** Physical and mechanical parameters of the coal and rock.

| no. | lithology | thickness (m) | density (kg m$^{-3}$) | bulk modulus (GPa) | shear modulus (GPa) | cohesion (MPa) | internal friction angle (°) | tensile strength (MPa) |
|---|---|---|---|---|---|---|---|---|
| 1 | aeolian sand | 20.0 | 2200 | 0.5 | 0.3 | 0.8 | 10 | 0.5 |
| 2 | sandy mudstone and sandstone interbed | 26.5 | 2400 | 6.7 | 2.7 | 2.9 | 28 | 1.3 |
| 3 | 2# coal seam | 3.5 | 1400 | 1.2 | 0.7 | 1.1 (0.11)[a] | 30 (20)[a] | 1.0 |
| 4 | sandy mudstone | 20.0 | 2450 | 9.6 | 4.4 | 3.5 | 29 | 2.3 |

[a]Numbers in the parentheses are residual values.

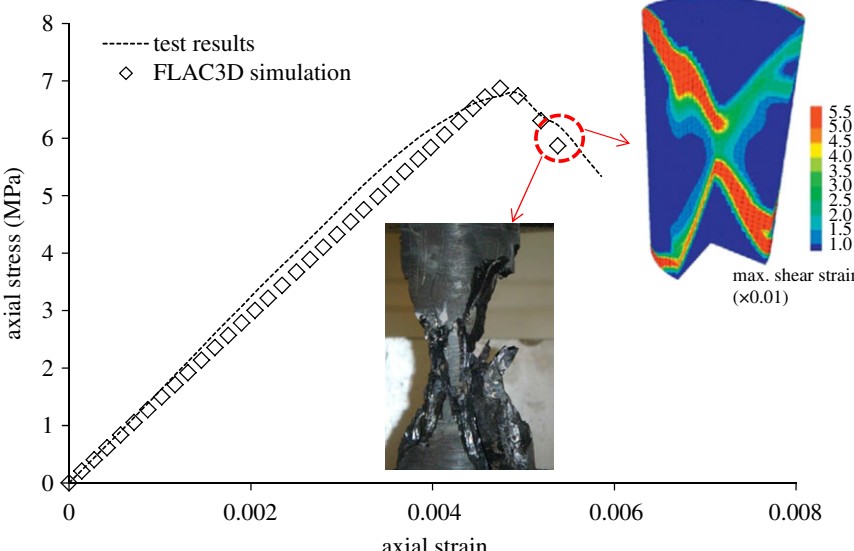

**Figure 4.** Stress–strain curves and failure mode of the laboratory test and numerical simulation in the coal UCS test.

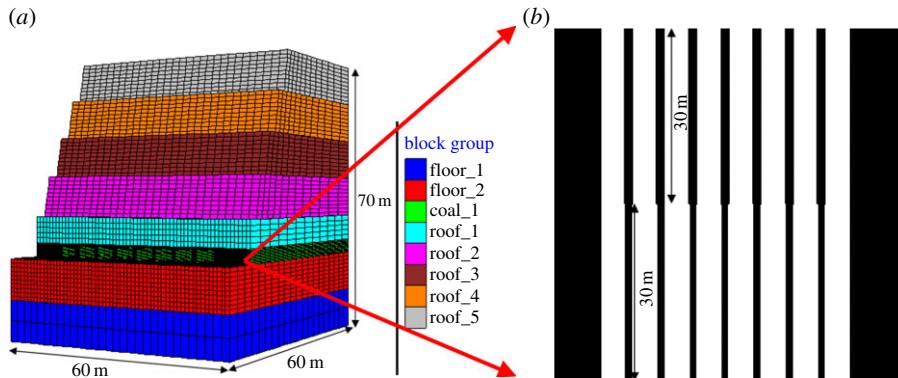

**Figure 5.** Three-dimensional model and sectional view of the coal seam. (*a*) Simulation model and (*b*) coal seam cross profile.

## 3.2. Empirical analysis

Underground pillars are mostly square and rectangular, whereas highwall mining pillars are long and narrow because they are formed after driving parallel entries in the seam from the highwall. These pillars are called web pillars. Several empirical coal pillar strength equations, which were developed for rectangular pillars, are modified for use with web pillars. However, for the rectangle pillar with a large aspect ratio, practice shows that the Mark-Bieniawski formula is the most suitable formula [27]:

$$S_{\mathrm{P}} = S_{\mathrm{I}}(0.64 + 0.36W/H), \tag{3.1}$$

where $S_{\mathrm{P}}$ is the coal pillar strength, MPa; $S_{\mathrm{I}}$ is the *in situ* coal strength, MPa; $W$ is the coal pillar width, m; $H$ is the mining height, m.

The tributary area method is useful for estimating the vertical stress on web and barrier pillars. The average vertical stress on the pillar is [28]

$$S_{\mathrm{WP}} = \frac{S_{\mathrm{V}}(W_{\mathrm{WP}} + W_{\mathrm{E}})}{W_{\mathrm{WP}}}, \tag{3.2}$$

where $S_{\mathrm{V}}$ is the *in situ* vertical stress, MPa; $W_{\mathrm{WP}}$ is the room coal pillar width, m; $W_{\mathrm{E}}$ is the highwall miner hole width, m.

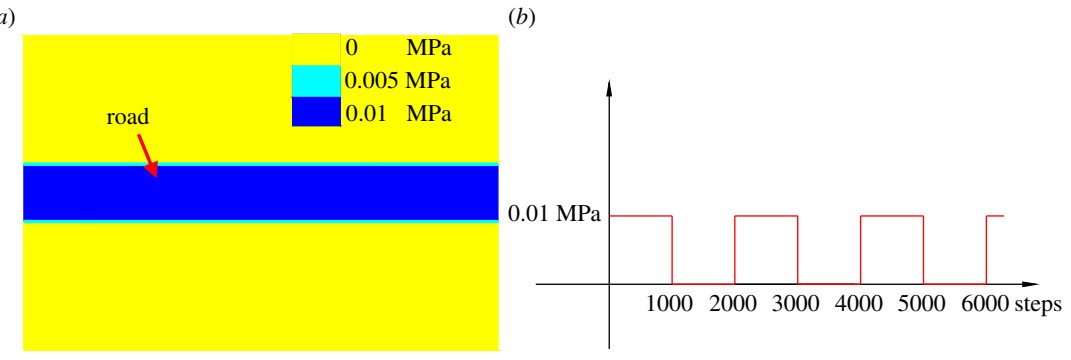

**Figure 6.** Dynamic load applied on the road. (*a*) Dynamic load on the street and (*b*) dynamic load path.

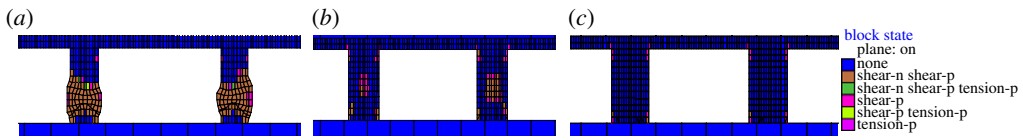

**Figure 7.** Plastic zone of the coal pillar under the road outside. (*a*) 1.0 m, (*b*) 1.3 m, and (*c*) 1.5 m.

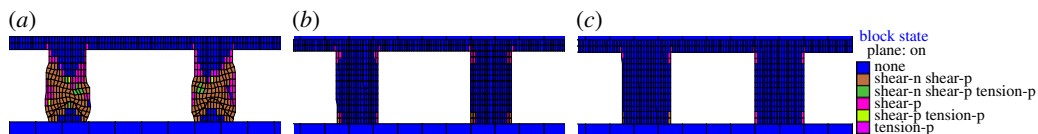

**Figure 8.** Plastic zone of the coal pillar under the road. (*a*) 1.4 m, (*b*) 1.7 m, and (*c*) 2.0 m.

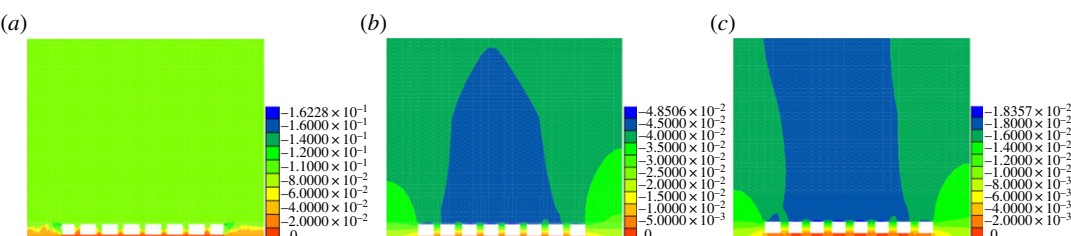

**Figure 9.** Vertical displacement of the road section with different pillar widths. (*a*) 1.4 m wide coal pillar, (*b*) 1.7 m wide coal pillar, and (*c*) 2.0 m wide coal pillar.

The overburden depth may be taken as the maximum overburden depth on a highwall mining web pillar, which is notably conservative, or as a high average value as follows [25]:

$$D_{\text{Design}} = 0.75 \times D_{\text{Max}} + 0.25 \times D_{\text{Min}}, \tag{3.3}$$

where $D_{\text{Max}}$ is the maximum overburden depth, m; $D_{\text{Min}}$ is the minimum overburden depth, m.

Neglecting the stress carried by the pillars (i.e. assuming that they have all failed), we obtain the average vertical stress on a barrier pillar [29].

$$S_{\text{BP}} = \frac{S_{\text{V}}(W_{\text{PN}} + W_{\text{BP}})}{W_{\text{BP}}}, \tag{3.4}$$

where $W_{\text{PN}}$ is the panel width, m; $W_{\text{BP}}$ is the barrier pillar width, m.

According to the numerical simulation result, the width of the barrier coal pillar $W_{\text{BP}}$ is 8 m, the width of the web coal pillar under the road $W_{\text{WPI}}$ is 1.7 m, and the width outside the road $W_{\text{WPO}}$ is 1.3 m. The depths of the coal seam inside and outside the road were calculated using equation (3.3) ($D_{\text{DesignI}} = 60$ m, $D_{\text{DesignO}} = 45$ m). The panel width $W_{\text{PN}}$ is 60 m, and the rock density $r$ is 24 000 N m$^{-3}$, so the *in situ* vertical stress $S_{\text{VI}}$ is 1.44 MPa, and $S_{\text{VO}}$ is 1.08 MPa. The strength of the *in situ* coal is 6.89 MPa

according to the mechanics experiment in the laboratory. Therefore, the strength of the barrier pillar is 12.91 MPa, the strength of the coal pillar under the road is 5.82 MPa, and the strength of the coal pillar outside the road is 5.48 MPa. The vertical stresses of the coal seam inside and outside the road are 4.7 MPa and 4.3 MPa according to equation (3.2), and $S_{BP}$ is 12.24 MPa according to equation (3.4), all of which are less than the corresponding strength of the coal pillar calculated according to equation (3.1) and similar to the strength of the coal pillar. The result shows that the width of the coal pillar obtained from the numerical simulation can satisfy the support requirement and recover more resources. However, it also shows that the empirical formulae have a large surplus coefficient, and the effect of the dynamic load of the road is not considered. Thus, although the pillar width determination method in this paper cannot be directly applied to other geological conditions, the pillar width of other geological conditions can be obtained using the method in this paper. Moreover, the numerical simulation can be used to obtain the reasonable width coal pillar with different road cross-section widths, pillar strengths, and overburden strata thicknesses. Thus, we can obtain a normalized empirical formula for the optimal width of coal pillars by setting a safety factor according to the numerical simulation.

According to the numerical simulation and theoretical analysis results, the design width of the coal pillar in the highwall is shown in figure 10.

# 4. Roadway layout and mining design

## 4.1. Roadway layout

For the highwall mining working faces, the gateway and pillar method is used to exploit the 2# coal seam with the characteristics of the roadway cross-section in figure 11.

In normal conditions, the roadway is 4.2 m wide, but it becomes 3.8 m wide when it develops under the road, and the heights are both 3.0 m. Because the cantilever excavator is remote controlled to mine, there is no need to excavate the roadway for humans. Although there is no permanent support in the roadway, the 0.5 m upper coal is treated as the temporary support to prevent the roof fall from damaging the devices. On the roadway, the protective shed is set before driving. The protective sheds are made from #16 joist steel with a 6-mm steel plate, and the support is made of a 114 mm steel tube. There are 8 sheds in total, which are 4.5 m long, 1.5 m wide and 3.3–3.7 m high.

## 4.2. Mining design

The HTM working face is located at the northwestern boundary of the coal mine, which is seated in the 2# coal seam with 390 m along the strike direction and 60 m along the dip direction. The mining design is shown in figure 12. This area cannot be exploited by the open-pit mining method. To improve the recovery ratio, the coal mine purchases an EBH132 cantilever excavator, which was produced and developed by NHIG, to perform the highwall mining, as shown in figure 12. After the coal resource in the end slope has been mined, the inner spoil dump backfills in a timely manner.

The EBH132 cantilever excavator is used for the mining and coaling, and the 650 belt conveyer is used for transportation. The main characteristics and working principles of the EBH-132 cantilever excavator are as follows:

— *Man−machine separation and high safety*. The system consists of a cantilever excavator and a remote control room, which is comprised of a KXB1-200/1140 (660) ZE mining flameproof electrical control box, a cantilever excavator electrical control box and a control panel. The system has the remote cable control function and achieves man−machine separation. When mining, the workers can operate the machine in the control room, as shown in figure 13. The cantilever excavator enters the highwall working face to finish mining to reduce the possibility of casualties and achieve safe mining.
— *Visualized operations and remote real-time control cantilever excavator*. Three cameras are installed in the cantilever excavator (the back one is added by the coal mine, as shown in figure 13). Using video imaging, the operator can see the projection of the seam−rock interface and guide the progress of the continuous miner. The work image of the cantilever excavator is transmitted to the control room through the signal transmission equipment, which achieves a real-time display of the working condition of each system using the screen on the control panel. It can satisfy

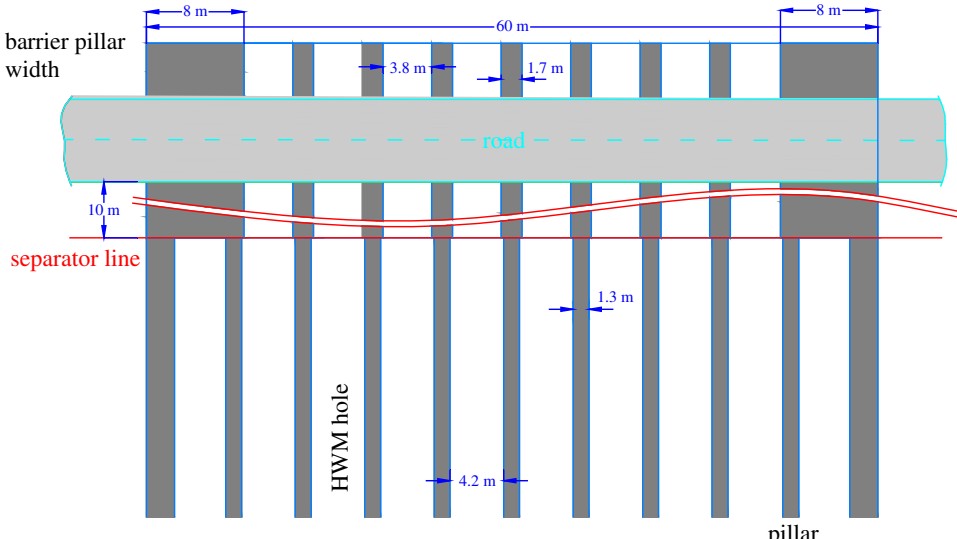

**Figure 10.** Highwall coal pillar width design.

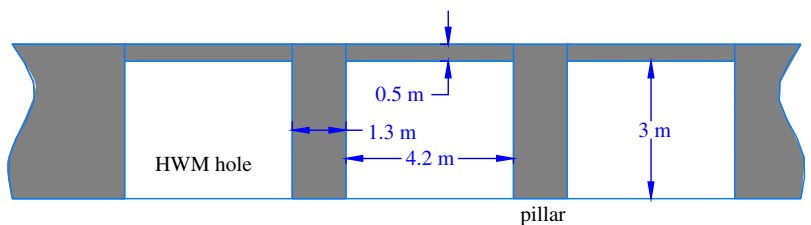

**Figure 11.** Excavation and residual pillar width design.

 the requirement for the controlling protection, display, illuminating and signal receiving functions of the cantilever excavator.

— *Simple mining process*. The main process is: preparation work → cutting → coaling → installing the belt conveyer → transporting by belt conveyer → casting by loader → digging to the design location and coming out of the roadway → equipment retracement → overhauling, as shown in figure 13. In the process of cutting, the first step is to adjust the cutting head to the right bottom corner and cut into the coal body. The cutting sequence is from right to left and subsequently from the bottom to the upper position. When the cutting head reaches the roof, the cantilever excavator goes backward to cut the top coal, which makes the roof smooth. When the process completes, we adjust the cutting head to the bottom and start another circulation. The details are shown in figure 14.

# 5. Field application effect analyses

The EBH132 cantilever excavator is used to exploit the end-wall residual coal resource in open-pit coal mines, and in comparison to other methods it is characterized by a high degree of mechanization, fewer workers, less labour density, good working environment and high efficiency. Regarding the work mode, two driving teams are required every day, and every team is composed of a team leader, a machine unit driver, a belt conveyer driver, a loading-machine runner and an electrician, totalling 10 workers who are required every day. After the team finishes mining the designed roadway and draws back the equipment, the machine should be overhauled. The length of driving is 10 m every day, and the work efficiency of each worker is 17.6 tons per day.

 Within two years of mining time, 710 m of the 2# coal seam was excavated in the strike direction, the coal reserves were 435 thousand tons, and the recoverable reserves were 291 thousand tons, except for the security coal pillar of 10-kVA high-voltage power lines and 35-kVA electrical substation. The highwall mining produced 197 thousand tons of coal, and the recovery ratio was above 67%.

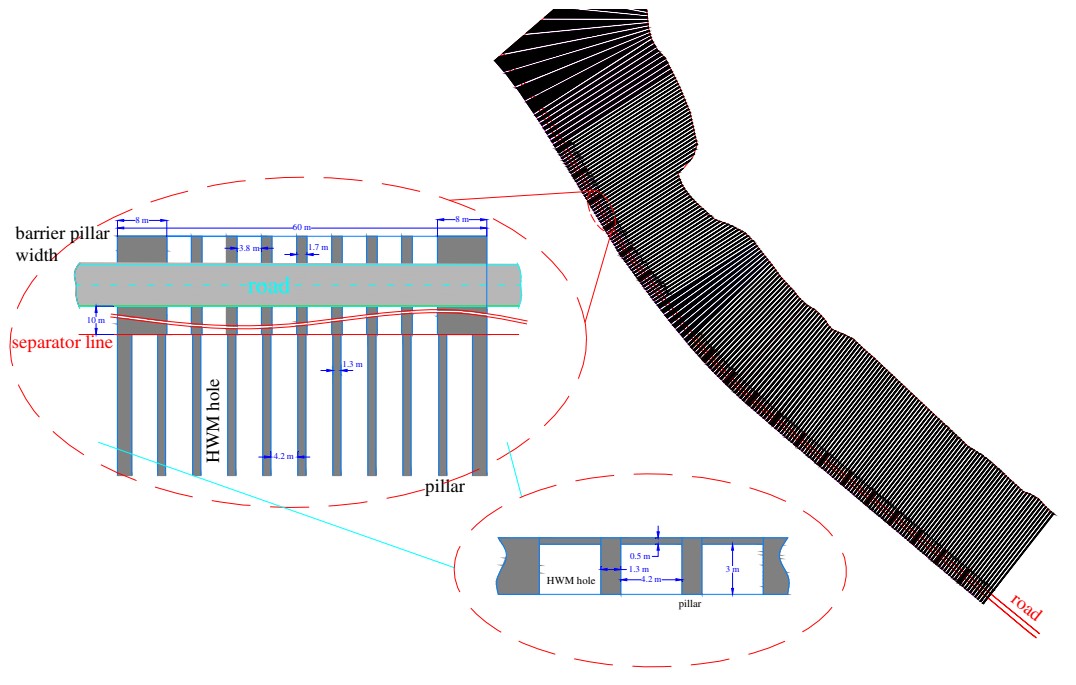

**Figure 12.** Highwall mining design planning map.

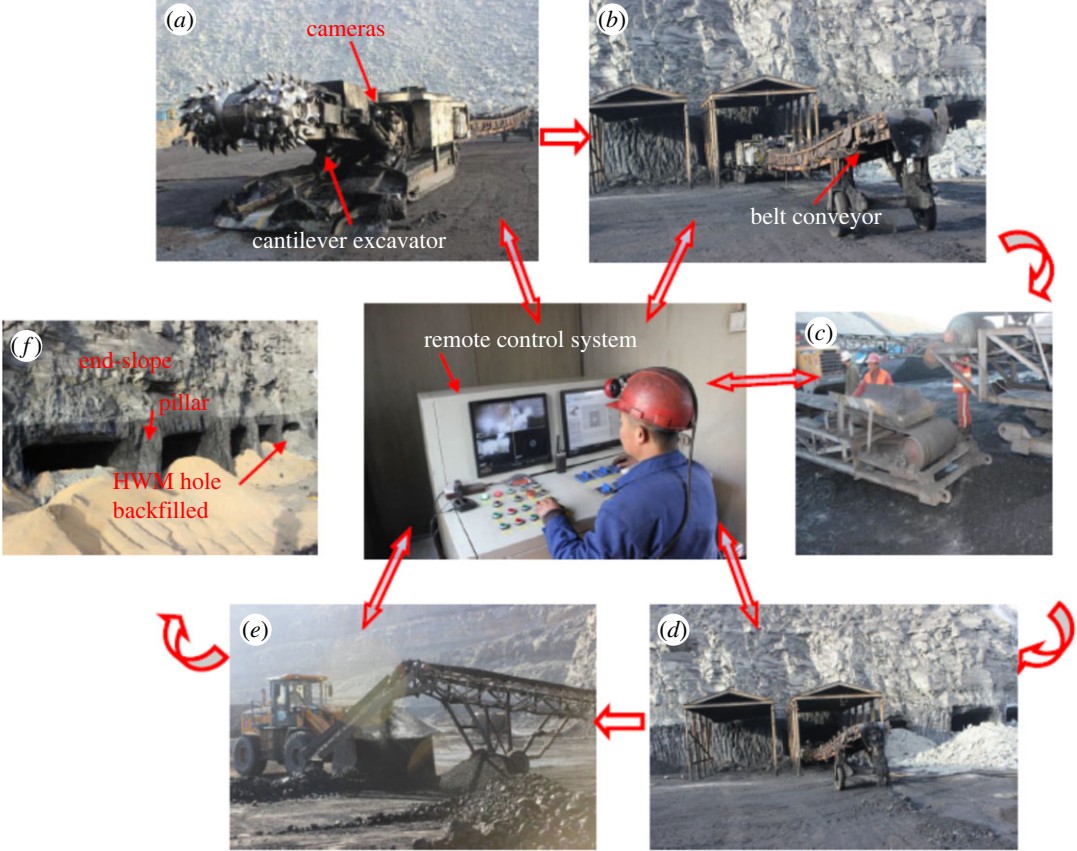

**Figure 13.** Simple mining process. (*a*) Preparatory work, (*b*) mining and loading, (*c*) installing the belt conveyer, (*d*) transporting by belt conveyer, (*e*) casting by loader and (*f*) HWM hole backfilled.

To further verify the stability of the selected coal pillars, the roadway section scanning analyses were carried out in the roadway under the road after highwall mining. The test method was shown in figure 15*a*. However, due to the fact that the HWM hole will be backfilled after highwall mining

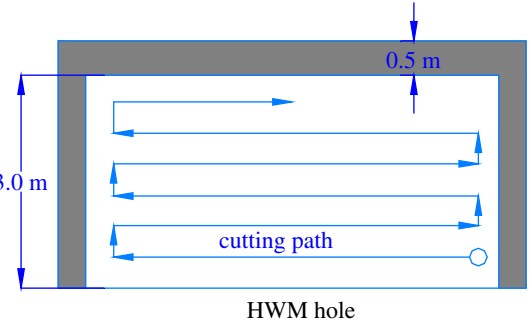

**Figure 14.** Excavator cutting path.

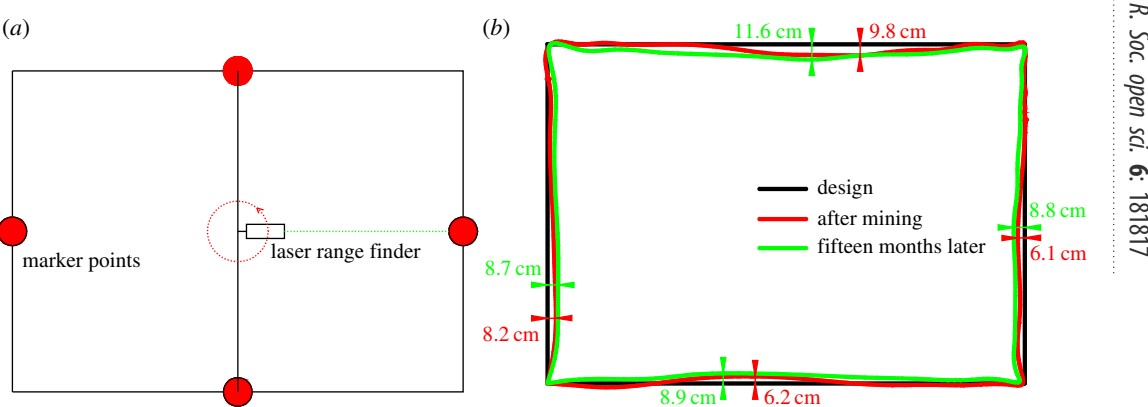

**Figure 15.** (*a*) The monitoring method and (*b*) the deformation monitoring results of the exposed roadway section.

(figure 13*f*), it is impossible to put the equipment into the roadway for measurement. Therefore, we can only monitor the deformation of the exposed roadway section; the monitoring results are shown in figure 15*b*. Within 15 months of mining, the maximum deformation of the roadway section is only 2.7 cm, which demonstrates that the design width of the coal pillar is sufficient to achieve reasonable stability.

# 6. Conclusion

To ensure the stability of coal pillars and prevent rock fall accidents in the process of HWM, the paper calculated and analysed the reasonable width of the coal pillar using numerical simulations and the strength theory. Then, the reasonable coal pillar widths outside and under the road were determined to be 1.7 m and 1.3 m, respectively, which can efficiently ensure the safety of ground facilities.

The EBH132 cantilever excavator has many advantages such as man–machine separation visualized operations and remote real-time control, which can achieve highwall mining with remote control. Hence, the safe, efficient and high-recovery-ratio highwall mining was achieved for the residual coal resources of an open-cast mine.

Regarding the open-cast mine, the remote control cantilever excavator was used to develop the unmanned highwall mining, which created a better working environment and a higher efficiency for the single worker. The recovery ratio was over 67%, and significant technical and economic results were achieved. During the mining process, there was no collapse of the coal pillars or roof caving accident, which demonstrates that the design width of coal pillar is reasonable, and the use of the residual coal resource in the end-wall area is efficient for the open-pit coal mine.

Data accessibility. Our data are from the field case. The numerical parameters are based on the geological data of a coal reservoir in Inner Mongolia Autonomous Province, China, which are listed in table 1. The data of the field application effect are described in detail in the field application section (§5) of this manuscript.
Authors' contributions. F.W. performed the mining parameter design, participated in the field test, and drafted the manuscript. C.Z. conducted the numerical simulation and conceived of, designed, and coordinated the study. All authors gave their final approval for publication.

Competing interests. We declare we have no competing interests.

Funding. Financial support came from the Fundamental Research Funds for the Central Universities (grant no. 2017CXNL01).

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
