## [Reviewer comments · Royal Society Open Science]

Review History

RSOS-181817.R0 (Original submission)

Review form: Reviewer 1 (Nicholas Fantuzzi)

Is the manuscript scientifically sound in its present form?

Yes

Are the interpretations and conclusions justified by the results?

Yes

Is the language acceptable?

Yes

Is it clear how to access all supporting data?

Not Applicable

Do you have any ethical concerns with this paper?

No

Have you any concerns about statistical analyses in this paper?

No

Recommendation?

Accept with minor revision (please list in comments)

Comments to the Author(s)

Please see attached file (Appendix A).

Review form: Reviewer 2

Is the manuscript scientifically sound in its present form?

Yes

Are the interpretations and conclusions justified by the results?

Yes

Is the language acceptable?

Yes

Is it clear how to access all supporting data?

Not Applicable

Do you have any ethical concerns with this paper?

No

Have you any concerns about statistical analyses in this paper?

No

Recommendation?

Accept with minor revision (please list in comments)

Comments to the Author(s)

This is a well-organized paper; the highlight of this work is that the reasonable coal pillar width outside and under the road was quantitatively designed by Mark-Bieniawski formula and FLAC3D numerical simulation. The remote control mining technology for the highwall residual coal resources was proposed in this paper. It is a novel method for the exploitation of open-pit mine, outcrop in hilly terrain and shallow depth coal seams in Chinese coal mines. Thus, I recommend it to be published in Royal Society Open Science with some suggestion about language that might further improve the clarity of the manuscript. Here are some of the details:

(1) The English needs further substantial editing. A native or more fluent English speaker familiar with the material should be asked for help.

(2) below figure 4: 0.01 MPa intervals of 1000 steps.

Please, explain why you adopt this value, if this follows some norm for road or bridge design rules?

(3) what are the safety conditions or the stability criteria that pillars have to satisfy? the plastic

zones are all decreased and plastic zone linking phenomenon do not appear either, which deduces that both can keep the stability.

(4) Please clarify what is meant with instability as it is used through the whole document
In figure 8a, you say that the shrinkage reaches 16cm but from the figure it seems that this is not true as the no blue contours are visible.

(5) The result of simulation indicates that the stability would be much better when the width of pillar under the road and under the road outside is 1.3m and 1.7m respectively
again this seems wrong, according with figure 6 and 7 is exactly inverse.

(6) Section 3.2: The vertical stresses of coal seam inside and outside the road are 4.7 MPa and 4.3 MPa separately according to the equation 2 and the SBP is 12.24 MPa according to the equation 4, which are all less than the corresponding strengths of coal pillar calculated according to the equation 1 and close to the strength of coal pillar.

Review form: Reviewer 3

Is the manuscript scientifically sound in its present form?

No

Are the interpretations and conclusions justified by the results?

No

Is the language acceptable?

No

Is it clear how to access all supporting data?

Yes

Do you have any ethical concerns with this paper?

No

Have you any concerns about statistical analyses in this paper?

No

Recommendation?

Reject

Comments to the Author(s)

In the current manuscript, the pillar width in highwall mining and remote control of mining equipment have been studied. A modelling based on FLAC3D was conducted to analyze the pillar stability with different pillar width.

1. The English needs substantial editing. It is understandable and confusing. There are too many grammar mistakes. A native or more fluent English speaker familiar with the material should be asked for help.
2. It looks like a simple mixture of two parts, pillar width design and remote control. This manuscript is more like an engineering report rather than a research study, especially for the part of remote control.
3. Abstract:

- Novel?? It is really a simple and common mining method.
- A little wordy for the first sentence.
- 4. Part 1 "Introduction":
 - Poor. Because it does not consider many contributions in this field. Actually, many researchers conducted various studies on pillar stability and highwall mining, especially in USA.
- 5. Part 3
 - The title of 3.1.1 are not good. There is no result in this part.
 - Does Fig.3 fit your situation?
 - In your modelling, you set the overlying strata of coal seam as interbedded stratum, is that proper?
 - The excavate width is also one of most important factor affecting the pillar stability, but we do not see the number until Part 4. Why not study this factor?
 - In my opinion, the pillar of 1.7 or 1.4 m may be a yield pillar. I do not think the modelling results can reveal the true status.
 - Do the formulas you selected fit your situation? As I know, some of them are developed for yield pillar design, some of them are for barrier pillar.
- 6. Part 4
 - The part of remote control looks like needless.
- 7. Part 5
 - There is no data on pillar deformation or roadway shrinkage.
 - As a research paper, this part is not a good expression for field application.
- 8. References: too many references in Chinese language.

Decision letter (RSOS-181817.R0)

30-Nov-2018

Dear Dr Zhang,

The editors assigned to your paper ("Coal pillar design and remote control mining technology for the highwall residual coal resources") have now received comments from reviewers. We would like you to revise your paper in accordance with the referee and Associate Editor suggestions which can be found below (not including confidential reports to the Editor). Please note this decision does not guarantee eventual acceptance.

Please submit a copy of your revised paper before 23-Dec-2018. Please note that the revision deadline will expire at 00.00am on this date. If we do not hear from you within this time then it will be assumed that the paper has been withdrawn. In exceptional circumstances, extensions may be possible if agreed with the Editorial Office in advance. We do not allow multiple rounds of revision so we urge you to make every effort to fully address all of the comments at this stage. If deemed necessary by the Editors, your manuscript will be sent back to one or more of the original reviewers for assessment. If the original reviewers are not available, we may invite new reviewers.

When submitting your revised manuscript, you must respond to the comments made by the

referees and upload a file "Response to Referees" in "Section 6 - File Upload". Please use this to document how you have responded to the comments, and the adjustments you have made. In order to expedite the processing of the revised manuscript, please be as specific as possible in your response.

- Data accessibility

If you wish to submit your supporting data or code to Dryad (<http://datadryad.org/>), or modify your current submission to dryad, please use the following link:
<http://datadryad.org/submit?journalID=RSOS&manu=RSOS-181817>

- Competing interests

- Authors' contributions

- Acknowledgements

- Funding statement

Please note that Royal Society Open Science charge article processing charges for all new submissions that are accepted for publication. Charges will also apply to papers transferred to Royal Society Open Science from other Royal Society Publishing journals, as well as papers submitted as part of our collaboration with the Royal Society of Chemistry (<http://rsos.royalsocietypublishing.org/chemistry>). If your manuscript is newly submitted and subsequently accepted for publication, you will be asked to pay the article processing charge, unless you request a waiver and this is approved by Royal Society Publishing. You can find out more about the charges at <http://rsos.royalsocietypublishing.org/page/charges>. Should you have any queries, please contact openscience@royalsociety.org.

on behalf of Professor R. Kerry Rowe (Subject Editor)
openscience@royalsociety.org

Associate Editor's comments:

We have received three reviews of your manuscript, which has resulted in a split decision. Two of the reviewers are broadly positive towards your work, but recommend a number of changes need to be made prior to consideration for acceptance, while the third recommends the paper should not be published. Given the split nature of the recommendation, we are opting to give you the benefit of the doubt and allow a revision to be submitted. You should ensure that you fully incorporate the changes requested by the referees, or provide a reasoned rebuttal to explain why you have not done so, and you should also seek the advice of a language polishing service before resubmitting. Examples of services to assist in editing the manuscript may be found at <https://royalsociety.org/journals/authors/language-polishing/>. You should provide evidence of having your manuscript revised by such a service, in the form of a certificate of completion or similar.

If you are unable to fully satisfy the reviewers that your changes have improved the manuscript to a publishable standard after revision, you may not be granted a second opportunity to do so. We wish you every success in preparing a revision.

Comments to Author:

Reviewers' Comments to Author:

Reviewer: 1

Comments to the Author(s)

Please see attached file

Reviewer: 2

Comments to the Author(s)

This is a well-organized paper; the highlight of this work is that the reasonable coal pillar width outside and under the road was quantitatively designed by Mark-Bieniawski formula and FLAC3D numerical simulation. The remote control mining technology for the highwall residual coal resources was proposed in this paper. It is a novel method for the exploitation of open-pit mine, outcrop in hilly terrain and shallow depth coal seams in Chinese coal mines. Thus, I recommend it to be published in Royal Society Open Science with some suggestion about language that might further improve the clarity of the manuscript. Here are some of the details:

(1) The English needs further substantial editing. A native or more fluent English speaker familiar with the material should be asked for help.

(2) below figure 4: 0.01 MPa intervals of 1000 steps.

Please, explain why you adopt this value, if this follows some norm for road or bridge design rules?

(3) what are the safety conditions or the stability criteria that pillars have to satisfy? the plastic zones are all decreased and plastic zone linking phenomenon do not appear either, which deduces that both can keep the stability.

(4) Please clarify what is meant with instability as it is used through the whole document. In figure 8a, you say that the shrinkage reaches 16cm but from the figure it seems that this is not true as the no blue contours are visible.

(5) The result of simulation indicates that the stability would be much better when the width of pillar under the road and under the road outside is 1.3m and 1.7m respectively. Again this seems wrong, according with figure 6 and 7 is exactly inverse.

(6) Section 3.2: The vertical stresses of coal seam inside and outside the road are 4.7 MPa and 4.3 MPa separately according to the equation 2 and the SBP is 12.24 MPa according to the equation 4, which are all less than the corresponding strengths of coal pillar calculated according to the equation 1 and close to the strength of coal pillar.

Reviewer: 3

Comments to the Author(s)

In the current manuscript, the pillar width in highwall mining and remote control of mining equipment have been studied. A modelling based on FLAC3D was conducted to analyze the pillar stability with different pillar width.

1. The English needs substantial editing. It is understandable and confusing. There are too many grammar mistakes. A native or more fluent English speaker familiar with the material should be asked for help.

2. It looks like a simple mixture of two parts, pillar width design and remote control. This manuscript is more like an engineering report rather than a research study, especially for the part of remote control.

3. Abstract:

→ Novel?? It is really a simple and common mining method.

→ A little wordy for the first sentence.

4. Part 1 "Introduction":

→ Poor. Because it does not consider many contributions in this field.

Actually, many researchers conducted various studies on pillar stability and highwall mining, especially in USA .

5. Part 3

→ The title of 3.1.1 are not good. There is no result in this part.

→ Does Fig.3 fit your situation?

→ In your modelling, you set the overlying strata of coal seam as interbed stratum, is that proper?

→ The excavate width is also one of most important factor affecting the pillar stability, but we do not see the number until Part 4. Why not study this factor?

→ In my opinion, the pillar of 1.7 or 1.4 m may be a yield pillar. I do not think the modelling results can reveal the true status.

→ Do the formulas you selected fit your situation? As I know, some of them are developed for yield pillar design, some of them are for barrier pillar.

6. Part 4

→ The part of remote control looks like needless.

7. Part 5

→ There is no data on pillar deformation or roadway shrinkage.

→ As a research paper, this part is not a good expression for field application.

8. References: too many references in Chinese language.

Author's Response to Decision Letter for (RSOS-181817.R0)

See Appendix B.

RSOS-181817.R1 (Revision)

Review form: Reviewer 1 (Nicholas Fantuzzi)

Is the manuscript scientifically sound in its present form?

Yes

Are the interpretations and conclusions justified by the results?

Yes

Is the language acceptable?

Yes

Is it clear how to access all supporting data?

Not Applicable

Do you have any ethical concerns with this paper?

No

Have you any concerns about statistical analyses in this paper?

I do not feel qualified to assess the statistics

Recommendation?

Accept as is

Comments to the Author(s)

Considering the comments given and the answers of the authors the manuscript can be accepted in the present form.

Review form: Reviewer 2**Is the manuscript scientifically sound in its present form?**

Yes

Are the interpretations and conclusions justified by the results?

Yes

Is the language acceptable?

Yes

Is it clear how to access all supporting data?

Not Applicable

Do you have any ethical concerns with this paper?

No

Have you any concerns about statistical analyses in this paper?

No

Recommendation?

Accept as is

Comments to the Author(s)

None

Review form: Reviewer 3**Is the manuscript scientifically sound in its present form?**

No

Are the interpretations and conclusions justified by the results?

No

Is the language acceptable?

Yes

Is it clear how to access all supporting data?

Yes

Do you have any ethical concerns with this paper?

No

Have you any concerns about statistical analyses in this paper?

No

Recommendation?

Major revision is needed (please make suggestions in comments)

Comments to the Author(s)

This paper refers two key issues in highwall mining, pillar design and remote control. Numerical modeling and Mark's formula were used for pillar design. The remote control process was introduced. The results provide a good case for highwall mining.

The following is a list of particular issues of concern:

1. page 2: "However, current studies rarely consider the effect of the roads on the highwall and pillar design. The effect of the coal trucks on the road on the stability of coal pillars was also of less consideration." It is really better if this paper consider the effects of these two factors in pillar design. But it still not clear in present paper how you consider the effects, especially in the theoretical analysis.
2. Page 5, Part 3.1.1: This paper proposes?
3. Page 5: Strain softening was used to evaluate coal seam failure in FLAC modelling. Please show the details of the parameters of coal seam, not only list in Table 1.
4. Page 7, Fig.7/8: The plastic zone in (b) and (c) are so small, which is out of my knowledge. And the of plastic zone in Fig.7(b) is located in the core of pillar, while those in Fig. 7 (c) and Fig.8(b)/(c) are located only at the corners, why?
5. Page 8, Please check the word "instability", a little confusing.
6. Page 8, The title of Part 3.2 may be better as empirical analysis. we do not see any theory in this part, except some empirical formulas.
7. Page 8, Eqs.(2) to (4): This paper develops?
8. Pages 8 and 9, How can we get the value of in situ strength SI ? according to the mechanics experiment in the laboratory? Is that in-situ strength?
9. Part 4: This paper shows what the remote control system is, but as an academic paper, we need to know how and why you select these equipments, or the determination of the cutting path, and so on.
10. Part 5: More data or evidence of pillar stability, highwall stability...

Decision letter (RSOS-181817.R1)

31-Jan-2019

Dear Dr Zhang:

Manuscript ID RSOS-181817.R1 entitled "Reasonable coal pillar design and remote control mining technology for highwall residual coal resources" which you submitted to Royal Society Open Science, has been reviewed. The comments of the reviewer(s) are included at the bottom of this letter.

Please submit a copy of your revised paper before 23-Feb-2019. Please note that the revision deadline will expire at 00.00am on this date. If we do not hear from you within this time then it

will be assumed that the paper has been withdrawn. In exceptional circumstances, extensions may be possible if agreed with the Editorial Office in advance. We do not allow multiple rounds of revision so we urge you to make every effort to fully address all of the comments at this stage. If deemed necessary by the Editors, your manuscript will be sent back to one or more of the original reviewers for assessment. If the original reviewers are not available we may invite new reviewers.

- Ethics statement

- Data accessibility

- Competing interests

- Authors' contributions

- Acknowledgements

- Funding statement

Kind regards,

on behalf of Professor R. Kerry Rowe (Subject Editor)

Associate Editor Comments to Author:

Generally, Royal Society Open Science does not permit multiple rounds of revision; however, given the positive views of two of the reviewers, an exception is being granted, as the authors appear to be making 'good faith' efforts to improve their manuscript.

You are being given the opportunity to incorporate the changes recommended by the remaining referee who has substantive comments. You **MUST** ensure that you meet their requirements, as the revision will be returned to them to assess. If you do not satisfy this reviewer that the paper is ready for publication, no further revisions will be entertained and the manuscript rejected.

Reviewer comments to Author:

Reviewer: 2

Comments to the Author(s)

None

Reviewer: 1

Comments to the Author(s)

Considering the comments given and the answers of the authors the manuscript can be accepted in the present form.

Reviewer: 3

Comments to the Author(s)

This paper refers two key issues in highwall mining, pillar design and remote control. Numerical modeling and Mark's formula were used for pillar design. The remote control process was introduced. The results provide a good case for highwall mining.

The following is a list of particular issues of concern:

1. page 2: "However, current studies rarely consider the effect of the roads on the highwall and pillar design. The effect of the coal trucks on the road on the stability of coal pillars was also of less consideration." It is really better if this paper consider the effects of these two factors in pillar design. But it still not clear in present paper how you consider the effects, especially in the theoretical analysis.
2. Page 5, Part 3.1.1: This paper proposes?
3. Page 5: Strain softening was used to evaluate coal seam failure in FLAC modelling. Please show the details of the parameters of coal seam, not only list in Table 1.
4. Page 7, Fig.7/8: The plastic zone in (b) and (c) are so small, which is out of my knowledge. And the of plastic zone in Fig.7(b) is located in the core of pillar, while those in Fig. 7 (c) and Fig.8(b)/(c) are located only at the corners, why?
5. Page 8, Please check the word "instability", a little confusing.
6. Page 8, The title of Part 3.2 may be better as empirical analysis. we do not see any theory in this part, except some empirical formulas.
7. Page 8, Eqs.(2) to (4): This paper develops?
8. Pages 8 and 9, How can we get the value of in situ strength SI ? according to the mechanics experiment in the laboratory? Is that in-situ strength?
9. Part 4: This paper shows what the remote control system is, but as an academic paper, we need to know how and why you select these equipments, or the determination of the cutting path, and so on.
10. Part 5: More data or evidence of pillar stability, highwall stability...

Author's Response to Decision Letter for (RSOS-181817.R1)

See Appendix C.

RSOS-181817.R2 (Revision)

Review form: Reviewer 3

Is the manuscript scientifically sound in its present form?

Yes

Are the interpretations and conclusions justified by the results?

Yes

Is the language acceptable?

Yes

Is it clear how to access all supporting data?

Yes

Do you have any ethical concerns with this paper?

No

Have you any concerns about statistical analyses in this paper?

No

Recommendation?

Accept as is

Comments to the Author(s)

The manuscript can be accepted in the present form.

Decision letter (RSOS-181817.R2)

07-Mar-2019

Dear Dr Zhang,

I am pleased to inform you that your manuscript entitled "Reasonable coal pillar design and remote control mining technology for highwall residual coal resources" is now accepted for publication in Royal Society Open Science.

Kind regards,

Andrew Dunn

on behalf of Prof R. Kerry Rowe (Subject Editor)

Reviewer comments to Author:

Reviewer: 3

Comments to the Author(s)

The manuscript can be accepted in the present form.

Appendix A

This paper aims at investigating the influence of mining parameters on stability of end-wall slopes in remote control mining technology. The width of coal pillars is designed using numerical methods so that the stability of roof is guaranteed, moreover; the roadway layout is designed so that the recovery ratio has increased. Nonetheless, the authors are recommended to take into account the following notes before publication.

- Abstract-line 38: To analyze?
- Line 47-page 3: TO is extra.
- Line 3-page 4: This paper proposes?
- Line 12-page 5: Does not?
- Line 40-page 5: While the coal seam was adopted by strain softening?
- Line 57-page 5: Parameters of all rocks?
- Line 43-page 6: It is more rational to use the term “plastic zone” rather than “plastic collapse”
- The optimal width of coal pillar could be expressed as a normalized parameter, as a ratio of road cross section width, as an example, so that the outcome of this research can be used for general cases.
- Line 22-page 9: There’s no need to the roadway?

Appendix B

Dear Editors and Reviewers:

The authors really appreciate your great kindness and good comments concerning our manuscript entitled 'Coal pillar design and remote control mining technology for the highwall residual coal resources (Manuscript ID RSOS-181817)'. We are very grateful to the reviewers for his very useful suggestions on how to make statements well in our paper. The revised paper has been rewritten and improved according to the suggestions of the reviewers. We hope it is satisfied. Revised portion are marked in blue in the paper. The main corrections in the paper and the responds to the reviewer's comments are as following:

Responds to the reviewers' comments:

Firstly, specific to the language problem, we have sought professional English editorial services (American Journal Experts) help to smooth the English of this paper aimed at the language problems. These changes of the language will not influence the content and framework of the paper. And here we did not list the changes. We are very sorry for more trouble to you and special thanks to you for your good comments.

Reviewer: 1

This paper aims at investigating the influence of mining parameters on stability of end-wall slopes in remote control mining technology. The width of coal pillars is designed using numerical methods so that the stability of roof is guaranteed, moreover; the roadway layout is designed so that the recovery ratio has increased. Nonetheless, the authors are recommended to take into account the following notes before publication.

- Abstract-line 38:** To analyze?
- Line 47-page 3: TO is extra.
- Line 3-page 4: This paper proposes?
- Line 12-page 5: Does not?
Line 57-page 5: Parameters of all rocks?
- Line 22-page 9: There's no need to the roadway?

Modifications: we are so sorry for our poor written, we have revised the above mistakes. Besides, specific to the language problem, we have sought professional English editorial services help to smooth the English of this paper aimed at the language problems. We are very sorry for more trouble to you and special thanks to you for your good comments.

□ **Line 40-page 5:** While the coal seam was adopted by strain softening?

Modifications: The strain softening model can well match the uniaxial compression strength (UCS) test of the laboratory test. The numerical model and the selected properties were calibrated through comparison of the coal UCS test. It can be seen in following figure that a good agreement was achieved between the numerical results and the laboratory test for both stress–strain curve and failure mode of the sample. The maximum shear strain in the numerical model shows an X-shaped failure mode of the sample, which was consistent with the laboratory test. Thus, we added this paragraph in section 3.1.2 as

“In this model, the rock stratum is represented by Mohr-Coulomb model, the coal seam is described by the strain softening model, and the physical and mechanical parameters of the coal and rock in the model are listed in Table 1. The numerical model and selected properties were calibrated through comparison of the coal uniaxial compressive strength test [26]. Figure 4 shows a good consistency between the numerical results and the laboratory test for the stress–strain curve and failure mode of the sample. The maximum shear strain in the numerical model shows an X-shaped failure mode of the sample, which is consistent with the laboratory test.”

Figure 4. Stress–strain curves and failure mode of the laboratory test and numerical simulation in the coal UCS test.

□ **Line 43-page 6:** It is more rational to use the term “plastic zone” rather than “plastic collapse”

Modifications: thank you for your good suggestion, the “plastic collapse” was revised as “plastic zone”.

□ **The optimal width** of coal pillar could be expressed as a normalized parameter, as a ratio of road cross section width, as an example, so that the outcome of this research can be used for general cases.

Modifications: Thank you for your good suggestion, it must be recognized that we should extend our strategy application. In order to prevent the overlying rock fall accidents, this paper uses Mark-Bieniawski formula and FLAC3D numerical simulation to analyze the reasonable coal pillar width outside and under the road. Thus, for other geological settings, it needs to perform the

numerical simulation again. This will reduce the applicability of the the outcome of our research. In fact, the practice shows that the Mark-Bieniawski formula is the most suitable formula for the coal pillar selection. But this method often has a large surplus coefficient and the effect of road's dynamic load is not considered. Thus, though the pillar width determination method in this paper can not be directly applied to other geological conditions, but the pillar width of other geological conditions can be also obtained by the method in this paper. Of course, we can use the numerical simulation to obtain the reasonable width coal pillar with different road cross section width, pillar strength, overburden strata thickness. So we can get a normalized empirical formula for the optimal width of coal pillar. Thus, considering the reviewer's good suggestions, we added a paragraph in section 3.2: as "The result shows that the width of the coal pillar obtained from the numerical simulation can satisfy the support requirement and recover more resources. However, it also shows that the empirical formulas have a large surplus coefficient, and the effect of the dynamic load of the road is not considered. Thus, although the pillar width determination method in this paper cannot be directly applied to other geological conditions, the pillar width of other geological conditions can be obtained using the method in this paper. Moreover, the numerical simulation can be used to obtain the reasonable width coal pillar with different road cross-section widths, pillar strengths, and overburden strata thicknesses. Thus, we can obtain a normalized empirical formula for the optimal width of coal pillars."

Also we revised the Title of this paper as "Reasonable coal pillar design and remote control mining technology for highwall residual coal resources"

Special thanks to you for your good comments.

Reviewer: 2

Comments to the Author(s)

This is a well-organized paper; the highlight of this work is that the reasonable coal pillar width outside and under the road was quantitatively designed by Mark-Bieniawski formula and FLAC3D numerical simulation. The remote control mining technology for the highwall residual coal resources was proposed in this paper. It is a novel method for the exploitation of open-pit mine, outcrop in hilly terrain and shallow depth coal seams in Chinese coal mines. Thus, I recommend it to be published in Royal Society Open Science with some suggestion about language that might further improve the clarity of the manuscript. Here are some of the details:

(1) The English needs further substantial editing. A native or more fluent English speaker familiar with the material should be asked for help.

Modifications: we have sought professional English editorial help to smooth the English of this paper aimed at the language problems. We are very sorry for more trouble to you and special thanks to you for your good comments.

(2) below figure 4: 0.01 MPa intervals of 1000 steps.

Please, explain why you adopt this value, if this follows some norm for road or bridge design rules?

Modifications: It is really true as reviewer suggested that this sentence is not clear. 0.01 MPa refers to the load of the coal trucks. The quality of a truck loaded with coal is about 50 t, and the bottom area of a truck is about 50 m², so the load applied on the road is 0.01 MPa. The intervals of 1000 steps mainly because the steps achieve dynamic balancing are about 1000.

In order to clarify this sentence, we added a sentence as ‘The load is assumed to be 0.01 MPa intervals of 1000 steps, since the load of a truck loaded with coal is approximately 0.01 MPa, and the dynamic balancing steps are 1000.’ behind this sentence.

(3) what are the safety conditions or the stability criteria that pillars have to satisfy? the plastic zones are all decreased and plastic zone linking phenomenon do not appear either, which deduces that both can keep the stability.

Modifications: It is really true as reviewer suggested that “safety conditions” is not clear. Mohr coulomb failure criteria are used in our numerical model. The more the plastic zone in pillar means the lower the bearing strength of the pillar. However, the plastic zone remain have the residual strength to support together with the elastic zone, as shown in Figure.3(b). When the plastic zone cut-through the pillar as shown in Figure.5(a) in the original manuscript, the plastic penetration area doesn't have enough strength to support the roadway, as shown in Figure.3(c). Thus, safety conditions of the pillar is that the plastic zones don't cut-through the pillar, as shown in Figure.3(b), is the safety conditions.

‘comparing the condition of 1.3m with 1.5m , the plastic zones are all decreased and plastic zone linking phenomenon do not appear either, which deduces that both can keep the stability’ is revised as ‘comparing the condition of 1.0m, the plastic zones of 1.3 m and 1.5 m are all decreased and the plastic penetration area does not appear either, which deduces that both can keep the stability’

In order to clarify the safety conditions in the revised paper, section 3.1.1 is used to illustrate the reasonable width of coal pillar in this manuscript as follows:

3.1.1. Working status of the coal pillar

The working status of the coal pillar mainly includes three different situations, as shown in Figure 3. Mohr-Coulomb failure criteria are used in the numerical model. A more plastic zone in the pillar indicates a lower bearing strength of the pillar. However, the remaining plastic zone has the residual strength to support the elastic zone, as shown in Figure 3(b). When the plastic zone cuts through the pillar, the plastic penetration area does not have sufficient strength to support the roadway, as shown in Figure 3(c). Thus, the safety conditions of the pillar are that the plastic zones do not cut through the pillar. Figures 3(a, b) satisfy the requirement. However, if the coal pillar width is too large as Figure 3(a) shows, it will waste the resource and reduce the resource recovery. In general, the reasonable width of

coal pillar is shown in Figure 3(b): exploit as much coal resource as possible under the safety condition of the pillars.

(a) Too large coal pillar width

(b) Appropriate coal pillar width

(c) Too small coal pillar width

width

I-fractured zone; II - plastic zone; III - elastic zone

Figure 3. Plastic zone with different coal pillar widths

(4) Please clarify what is meant with instability as it is used through the whole document

In figure 8a, you say that the shrinkage reaches 16cm but from the figure it seems that this is not true as the no blue contours are visible.

Modifications: The ‘instability’ in this paper means that the pillar beyond the reach of safety conditions. As illustrate the “safety conditions” in the previous question, I think the instability in this paper can also well explain.

In Figure 8a, the main shrinkage of the model is 12 cm. However, the maximal shrinkage did reaches 16cm, as shown in the following enlarged figure. The maximal shrinkage is located at the top of roadway.

Figure 8(a) Vertical displacement of the road section with a pillar width of 1.4m

(5) The result of simulation indicates that the stability would be much better when the width of pillar under the road and under the road outside is 1.3m and 1.7m respectively again this seems wrong, according with figure 6 and 7 is exactly inverse.

Modifications: We are very sorry for our ambiguous expression. According to Figure 7, 8 and 9 in revised paper, the stability would be much better when the width of pillar outside and under the road is 1.5m and 2m, respectively. However, this status is the first status of the three pillars working status, which has a best stability but wasting the resources. Thus, the reasonable width of coal pillar outside and under the road is 1.3 m and 1.7 m, respectively.

In order to well illustrate this sentence, the revised sentence is “**The simulation result indicates that the reasonable widths of the pillar outside and under the road are 1.3 m and 1.7 m, respectively, which can satisfy the requirement of mining safety and a high recovery ratio. Therefore, this condition was considered the reasonable width of a coal pillar after comprehensive comparison.**”

(6) Section 3.2: The vertical stresses of coal seam inside and outside the road are 4.7 MPa and 4.3 MPa separately according to the equation 2 and the SBP is 12.24 MPa according to the equation 4, which are all less than the corresponding strengths of coal pillar calculated according to the equation 1 and close to the strength of coal pillar.

Modifications: We are very sorry for our ambiguous expression. The main means of this sentence is that the coal pillar used in this manuscript is not too large to the field application just like the first status of the three working status. Thus, the rational calculation should be under the corresponding strengths of coal pillar but not difference too much.

Special thanks to you for your good comments.

Reviewer: 3

Comments to the Author(s)

In the current manuscript, the pillar width in highwall mining and remote control of mining equipment have been studied. A modelling based on FLAC3D was conducted to analyze the pillar stability with different pillar width.

1. The English needs substantial editing. It is understandable and confusing. There are too many grammar mistakes. A native or more fluent English speaker familiar with the material should be asked for help.

Modifications: We have sought professional English editorial help to smooth the English of this paper aimed at the language problems. We are very sorry for more trouble to you and special thanks to you for your good comments.

2. It looks like a simple mixture of two parts, pillar width design and remote control. This manuscript is more like an engineering report rather than a research study, especially for the part of remote control.

Modifications: Thank you for your good suggestion. It is really true as the reviewer's comments that the correlation between the pillar width design and remote control technology is not well expressed. Highwall mining is a technique for attaining additional coal recovery after the economic strip limit is reached in surface mining. It involves remote deployment of a continuous miner in openings beneath the final highwall, with no personnel entry. Thus, the stability of coal pillars is the precondition for safe implementation of remote control mining. Both of them constitute the highwall mining technology. The remote control technology determines the main roadway layout of highwall mining system. There is no need to the roadway by human, and the excavate width is determined by the size of EBH132 cantilever excavator. Moreover, there is no permanent support in roadway. Thus, the stability of overburden structure mainly depends on the stability of retained coal pillars. In summary, remote control mining technology simplified mining system to achieve efficient mining and reasonable pillar width ensures the safe mining and increases the recovery ratio.

In order to address this problem, we added a paragraph at the end of the section 2 as “**The main problem of the highwall mining technology is the layout of roadways. In this paper, remote control technology is used, and the excavate width is determined by the size of the EBH132 cantilever excavator. There is no need to extract the roadway for human, which can simplify the mining system to achieve efficient mining. However, there is no permanent support in the roadway for the remote control mining. Thus, the stability of the overburden structure mainly depends on the stability of the retained coal pillars, and a reasonable pillar width ensures the safe mining and increases the recovery ratio. Thus, the reasonable coal pillar design and remote control technology for the HWM technology were studied in this paper.**”

3. Abstract:

Novel?? It is really a simple and common mining method.

A little wordy for the first sentence.

Modifications: It is really true as reviewer's comment that highwall mining technology is a simple and common mining method in the world. Considering the reviewer's good suggestion we revised the first sentence as "Highwall mining (HWM) technology is an efficient method for exploiting residual coal resources in Chinese open-pit coal mines."

4. Part 1 "Introduction":

Poor. Because it does not consider many contributions in this field.

Actually, many researchers conducted various studies on pillar stability and highwall mining, especially in USA .

Modifications: Highwall mining is a simple and common mining method in the world and we revised the introduction of this part. Besides, we used English references, similar to original Chinese references content, to replace part of Chinese references. For pillar stability studies of highwall mining, we added a paragraph as:

A reasonable mining pillar for highwall mining is conducive to safe and efficient mining. The highwall mining pillar design is a direct function of the coal strength, opening height, opening width, and depth of cover. An elasto-plastic model suitable for the analysis of coal pillars has been developed and implemented in both two- and three-dimensional finite element codes by Fama et al. [20]. The use of the local mine stiffness concept can provide added confidence to a highwall mining panel layout design [21]. Web and barrier pillar recommendations for close-proximity multiple-seam highwall mining were studied by Mark [22, 23]. Perry et al. [24] studied the effect of the highwall mining progression on the web and barrier pillar stability. Using numerical modeling tools, a correction factor was suggested in the empirical pillar strength equation for slender pillars with width-to-height ratios less than unity [25]. However, current studies rarely consider the effect of the roads on the highwall and pillar design. The effect of the coal trucks on the road on the stability of coal pillars was also of less consideration.

5. Part 3

→ The title of 3.1.1 are not good. There is no result in this part.

→ Does Fig.3 fit your situation?

Modifications: It is really true that there is no result in section 3.1.1. The purpose of this passage is listing three working states of coal pillars. Thus, we revised the title of section 3.1.1 as "Working status of the coal pillar". Fig.3 shows the plastic zone with different coal pillar widths. It is mainly for illustration the reasonable width of coal pillar and showing the stress distribution in coal pillar. Considering the reviewer's good suggestion, we revised the Fig.3 as follows:

Figure 3. The plastic zone with different coal pillar widths

→ In your modelling, you set the overlying strata of coal seam as interbed stratum, is that proper?

Modifications: FLAC3D is simulation software based on finite difference method, so the whole stratum is still continuous. The interbed stratum is used for parameter assignment of the strata with different strength. We cleared it in the revised paper.

→ The excavate width is also one of most important factor affecting the pillar stability, but we do not see the number until Part 4. Why not study this factor?

Modifications: It is really true as as reviewer's comment that the excavate width is also one of most important factor affecting the pillar stability. Highwall mining pillar design is a direct function of coal strength, opening height, opening width, and depth of cover. Depth of cover and opening height is depending on geological conditions. Coal strength mainly depends on the strength of coal and the width of coal pillar. Thus, the coal pillar width and opening width are the main factors affecting the pillar stability. And in our situation, the excavate width is determined by the size of EBH132 cantilever excavator. And the total width of coal pillar width and opening width is fixed (5.5 m) before the coal pillar width design. Thus, in the numerical simulation, the excavate width is determined by the coal pillar width. The width of pillar under the road outside was simulated to be 1.0 m, 1.3 m and 1.5 m (the excavate width was simulated to be 4.5 m, 4.2 m and 4 m) and the width of pillar under the road was simulated to be 1.4 m, 1.7 m and 2.0 m (the excavate width was simulated to be 4.1 m, 3.8 m and 3.5 m), respectively. Considering the reviewer's good comments, we added the sentence in the section 3.1.2 as follows:

The pillar under the road outside was simulated to be 1.0 m, 1.3 m and 1.5 m wide, and the pillar under the road was simulated to be 1.4 m, 1.7 m and 2.0 m wide, respectively. According to the size of the residual coal and cantilever excavator, the width of each panel (sum of the pillar width and excavate width) was set to 5.5 m. Thus, the excavation width under the road outside was simulated to be 4.5 m, 4.2 m and 4 m, and the excavation width under the road was simulated to be 4.1 m, 3.8 m and 3.5 m, respectively.

→ In my opinion, the pillar of 1.7 or 1.4 m may be a yield pillar. I do not think the modelling results can reveal the true status.

Modifications: The numerical model was used to back-analyze the uniaxial strength test of the coal in the hope to find the parameter for yield pillar design. In order to improve the reliability of the numerical model, the strain-softening pillar materials were meticulously calibrated. In this model, the internal friction angle, cohesion and strength of extension will decrease with the strain increases. The numerical model and the selected properties were calibrated through comparison of the coal UCS test. It can be seen in following figure that a good agreement was achieved between the numerical results and the laboratory test for both stress–strain curve and failure mode of the sample. The maximum shear strain in the numerical model shows an X-shaped failure mode of the sample, which was consistent with the laboratory test. Thus, we think the modelling results can reveal the true status. Considering the reviewer’s good suggestion, we added this paragraph in section 3.1.2.

“In this model, the rock stratum is represented by Mohr-Coulomb model, the coal seam is described by the strain softening model, and the physical and mechanical parameters of the coal and rock in the model are listed in Table 1. The numerical model and selected properties were calibrated through comparison of the coal uniaxial compressive strength test [26]. Figure 4 shows a good consistency between the numerical results and the laboratory test for the stress–strain curve and failure mode of the sample. The maximum shear strain in the numerical model shows an X-shaped failure mode of the sample, which is consistent with the laboratory test.”

Figure 4. Stress–strain curves and failure mode of the laboratory test and numerical simulation in the coal UCS test.

Pillar design is of paramount importance to any underground mine design. Oversized pillars may lead to loss of coal while undersized pillars may lead to instability. In fact, the numerical results of the pillar of 1.4 m show that the plastic zone spreads throughout the whole coal pillar. Thus, the coal pillar is in yielding state, as shown in Figure 8 (a), the bearing capacity of coal pillar has declined significantly, causing the shrinkage to reach 16cm, which is far more than the others. When the width increases to 1.7m, the plastic zone is mainly distributed in the corners, but the scope is relatively small and it does not spread throughout the whole pillar. The bearing capacity can satisfy the requirement. Meanwhile, the maximum shrinkage is 4.8 cm in the cross section of

highway. Thus, it meets the engineering requirement.

Thus, the pillar of 1.4 m is a yield pillar (the third case in Figure 3), while the pillar of 1.7 m is only part of the boundary area yielded (the second case in Figure 3), it can bear the load of overlying strata.

→ Do the formulas you selected fit your situation? As I know, some of them are developed for yield pillar design, some of them are for barrier pillar.

Modifications: The width of yield pillar is the main research object in this paper. While underground pillars are mostly square and rectangular, highwall mining pillars are long and narrow, as they are formed after driving parallel entries in the seam from the highwall. These pillars are termed as web pillars. Several empirical coal pillar strength equations developed for rectangular pillars are still being used with some modifications to adapt to web pillars. However, as to the rectangle pillar with large aspect ratio, the practice shows that the Mark-Bieniawski formula is the most suitable formula for the yield pillar design. In order to clear this problem, we added a sentence in Section 3.2 and added the relevant references.

6. Part 4

→ The part of remote control looks like needless.

Modifications: the remote control is the part of highwall mining technology. This part can make the structure of this case study more reasonable. Also we revised the Title of this paper as “Reasonable coal pillar design and remote control mining technology for highwall residual coal resources”

7. Part 5

→ There is no data on pillar deformation or roadway shrinkage.

→ As a research paper, this part is not a good expression for field application.

Modifications: It is really true as reviewer’s comment that this part is not a good expression for field application. To further verify the stability of the selected coal pillars, the roadway section scanning analyses should be carried out in the roadway under the road after highwall mining. The test method was shown in the following figure. However, due to the HWM hole will be backfilled after highwall mining (Fig.14f), it is impossible to put the equipment into the roadway for measurement. Therefore, the deformation of roadways was not monitored in this paper. During the process of mining, there is neither collapse of coal pillars nor roof caving accident, which demonstrates that the design width of coal pillar is reasonable and the exploitation of the residual coal resource in the end-wall area is efficient for open-pit coal mine.

Figure Monitoring method of roadway based on laser ranging

8. References: too many references in Chinese language.

Modifications: Considering reviewer's good suggestion, we used English references, similar to original Chinese references content, to replace part of Chinese references. There are only three Chinese references in the total 27 new references.

Special thanks to you for your good comments.

We tried our best to improve the manuscript and made some changes in the manuscript. These changes will not influence the content and framework of the manuscript. And here we did not list the changes but marked in blue in the revised edition. We appreciate for Editors/Reviewers' warm work earnestly, and hope that the correction will meet with approval.

Once again, thank you very much for your comments and suggestions

Yours sincerely,

Cun Zhang

Appendix C

Dear Editors and Reviewers:

The authors really appreciate your great kindness and good comments concerning our manuscript entitled “Reasonable coal pillar design and remote control mining technology for highwall residual coal resources” (Manuscript ID RSOS-181817.R1). We are very grateful to the reviewer for his very useful suggestions on how to make statements well in our paper. The revised paper has been rewritten and improved according to the suggestions of the reviewer. We hope it is satisfied. The main corrections in the paper and the responds to the reviewer’s comments are as following:

Responds to the reviewer’s comments:

Reviewer: 3

This paper refers two key issues in highwall mining, pillar design and remote control. Numerical modeling and Mark’s formula were used for pillar design. The remote control process was introduced. The results provide a good case for highwall mining.

The following is a list of particular issues of concern:

1. page 2: “However, current studies rarely consider the effect of the roads on the highwall and pillar design. The effect of the coal trucks on the road on the stability of coal pillars was also of less consideration.” It is really better if this paper consider the effects of these two factors in pillar design. But it still not clear in present paper how you consider the effects, especially in the theoretical analysis.

Modifications: It is really true as reviewer’s comments that the theoretical analysis were not consider the effect of the roads. In this paper, to analyze the effect of the coal trucks on the road on the stability of the coal pillars, a dynamic load was applied in the numerical simulation. The theoretical analysis was used to validate the numerical simulation results. And the result shows that the width of the coal pillar obtained from the numerical simulation can satisfy the support requirement and recover more resources. However, it also shows that the empirical formulas have a large surplus coefficient due to the effect of the dynamic load of the road are not considered. Thus, although the pillar width determination method in this paper cannot be directly applied to other geological conditions, the pillar width of other geological conditions can be obtained using the method in this paper. Moreover, the numerical simulation can be used to obtain the reasonable width coal pillar with different road cross-section widths, pillar strengths, and overburden strata thicknesses. Thus, we can obtain a normalized empirical formula for the optimal width of coal pillars by setting a safety factor according the numerical simulation.

In order to clear this problem, we added a discussion for the theoretical analysis at the end of the section as:

The result shows that the width of the coal pillar obtained from the numerical simulation can satisfy the support requirement and recover more resources. However, it also shows that the empirical formulas have a large surplus coefficient, and the effect of the dynamic load of the road is not considered. Thus, although the pillar width determination method in this paper cannot be directly applied to other geological conditions, the pillar width of other geological conditions can be obtained using the method in this paper. Moreover, the numerical simulation can be used to obtain the reasonable width coal pillar with different road cross-section widths, pillar strengths, and overburden strata thicknesses. Thus, we can obtain a normalized empirical formula for the optimal width of coal pillars by setting a safety factor according the numerical simulation.

2. Page 5, Part 3.1.1: This paper proposes?

Modifications: We revised this sentence as “Thus, highwall mining technology was used in this paper to excavate the residual coal resource and improve the recovery rate.”

3. Page 5: Strain softening was used to evaluate coal seam failure in FLAC modelling. Please show the details

of the parameters of coal seam, not only list in Table 1.

Modifications: Thank you for your good suggestion, we added the clarification in Section 3.1.2 as **the coal seam is described by the strain softening model, the cohesion and friction angle of the coal seam degrades as the plastic shear strain increases, and these factors are assigned the residual values when the plastic shear strain reaches 0.01, the physical and mechanical parameters of the coal and rock in the model are listed in Table 1.**

Table 1. Physical and mechanical parameters of the coal and rock

No.	Lithology	Thickness/m	Density /kg·m ⁻³	Bulk modulus/GPa	Shear modulus /GPa	Cohesion /MPa	Internal friction angle/°	Tensile strength /MPa
1	Aeolian sand	20.0	2200	0.5	0.3	0.8	10	0.5
2	Sandy mudstone and sandstone interbed	26.5	2400	6.7	2.7	2.9	28	1.3
3	2# Coal seam	3.5	1400	1.2	0.7	1.1 (0.11) *	30 (20)*	1.0
4	Sandy mudstone	20.0	2450	9.6	4.4	3.5	29	2.3

* Numbers in the parentheses are residual values.

4. Page 7, Fig.7/8: The plastic zone in (b) and (c) are so small, which is out of my knowledge. And the of plastic zone in Fig.7(b) is located in the core of pillar, while those in Fig. 7 (c) and Fig.8(b)/(c) are located only at the corners, why?

Modifications: In general, the corners of the pillar are the stress concentration area, the plastic zone priority occurs in this area. Thus, Fig. 7 (c) and Fig.8(b)/(c) are located only at the corners shows the bearing capacity can satisfy the requirement. While in Fig.7(b), after the corners of the pillar yielded, the plastic zone then occurred in the core of pillar, it is due to the brittleness of coal, when the stress reaches a certain strength, tensile failure will occur in the middle part. When the plastic zone spreads throughout the coal pillar, the ultimate bearing capacity will significantly decline and make the coal pillar unstable. Thus, due to the plastic zone in Fig.7(b) were not developed throughout the coal pillar, it still maintain support strength. In order to clear this problem, we added a sentence in section 3.1.3.

As shown in Figure 7, when the coal pillar outside the highway is 1.0 m, the plastic zone has spread throughout the entire pillar, and the coal pillar collapses because of instability, which cannot satisfy the mining safety requirement; comparing the condition of 1.0 m, all plastic zones of 1.3 m and 1.5 m decreased. Due to the corners of the pillar are the stress concentration area, the plastic zone priority occurs in this area. Thus, Fig. 7 (c) located only at the corners shows the bearing capacity can satisfy the requirement. While in Fig.7(b), after the corners of the pillar yielded, the plastic zone then occurred in the core of pillar, it is due to the brittleness of coal, when the stress reaches a certain strength, tensile failure will occur in the middle part, but the plastic penetration area did not appear, so both zones maintained the stability. Therefore, the width of the coal pillar under the road outside is 1.3 m due to the high resource recovery ratio.

5. Page 8, Please check the word “instability”, a little confusing.

Modifications: instability was revised as “unstability”

6. Page 8, The title of Part 3.2 may be better as empirical analysis. we do not see any theory in this part, except some empirical formulas.

Modifications: The title of Part 3.2 were revised as “Empirical analysis”

7. Page 8, Eqs.(2) to (4): This paper develops?

Modifications: The references were cited for Eqs.(2) to (4).

8. Pages 8 and 9, How can we get the value of in situ strength SI ? according to the mechanics experiment in the laboratory? Is that in-situ strength?

Modifications: According to the mechanics experiment in the laboratory, the value of in situ strength S_I is calculated by the following formula, which can be seen from the reference [27]:

$$S_I = K / (36)^{1/2} (H > 0.9 \text{ m}) \text{ or } S_I = K / (H)^{1/2} (H < 0.9 \text{ m})$$

$$K = \sigma_c (D)^{1/2}$$

where, σ_c is the uniaxial compressive strength of coal specimens tested in laboratory, MPa; D is the diameter or cube side dimension, m; H is the pillar height.

9. Part 4: This paper shows what the remote control system is, but as an academic paper, we need to know how and why you select these equipments, or the determination of the cutting path, and so on.

Modifications: The selection of the remote control mining equipment mainly according to the following reasons:

- (1) Based on the field geological conditions and coal reserves of the highwall coal seam;
- (2) Due to the existing mining equipment research and development status, and the equipment improvement capability of the supplier.
- (3) In consideration of the safe mining in the roadway without permanent support.

As for the determination of the cutting path, the cutting sequence is from right to left and subsequently from the bottom to the upper position, because of the continuous and high-efficient working process, which has been extensively applied for the coal roadways excavation in the underground mines.

10. Part 5: More data or evidence of pillar stability, highwall stability...

Modifications: It is really true as reviewer's comment that there is little data for evidence of pillar stability or highwall stability. To further verify the stability of the selected coal pillars, the roadway section scanning analyses should be carried out in the roadway under the road after highwall mining. The test method was shown in the following figure. However, due to the HWM hole will be backfilled after highwall mining (Fig.14f), it is impossible to put the equipment into the roadway for measurement. Therefore, the deformation of roadways was not monitored in this paper.

In order to prove the stability of coal pillar as far as possible, we can only monitor the deformation of the exposed roadway section; the monitoring results are as follows. Within 15 months after mining, the maximum deformation of roadway section is only 2.7 cm, which demonstrates that the design width of coal pillar is reasonable and the exploitation of the residual coal resource in the end-wall area is efficient for open-pit coal mine.

(a)

(b)

Fig.15 the monitoring method (a) and the deformation monitoring results (b) of the exposed roadway section

Special thanks to you for your good comments.

We tried our best to improve the manuscript and made some changes in the manuscript. These changes will not influence the content and framework of the manuscript. And here we did not list the changes but marked in blue in the revised edition. We appreciate for Editors/Reviewers' warm work earnestly, and hope that the correction will meet with approval.

Once again, thank you very much for your comments and suggestions

Yours sincerely,

Cun Zhang